# Prediction of air quality in Sydney, Australia as a function of forest fire load and weather using Bayesian statistics

**Michael Anthony Storey** [1,2]*, Owen F. Price[1,2]

**1** NSW Bushfire Risk Management Research Hub, Wollongong, NSW, Australia, **2** Department of Earth, Atmospheric and Life Sciences, University of Wollongong, Wollongong, NSW, Australia

* mstorey@uow.edu.au

**Data Availability Statement:** VIIRS SNPP hotspots used for analysis are freely available via the NANA FIRMS website at https://firms.modaps.eosdis.nasa.gov/download/. New South Wales PM2.5 data

## Abstract

Smoke from Hazard Reduction Burns (HRBs) and wildfires contains pollutants that are harmful to human health. This includes particulate matter less than 2.5 μm in diameter ($PM_{2.5}$), which affects human cardiovascular and respiratory systems and can lead to increased hospitalisations and premature deaths. Better models are needed to predict $PM_{2.5}$ levels associated with HRBs so that agencies can properly assess smoke pollution risk and balance smoke risk with the wildfire mitigation benefits of HRBs. Given this need, our aim was to develop a probabilistic model of daily $PM_{2.5}$ using Bayesian regression. We focused on the region around Sydney, Australia, which regularly has hazard reduction burning, wildfires and associated smoke. We developed two regional models (mean daily and maximum daily) from observed $PM_{2.5}$, weather reanalysis and satellite fire hotspot data. The models predict that the worst $PM_{2.5}$ in Sydney occurs when $PM_{2.5}$ was high the previous day, there is low ventilation index (i.e. the product of wind speed and planetary boundary layer height), low temperature, west to northwest winds in the Blue Mountains, an afternoon sea breeze and large areas of HRBs are being conducted, particularly to the west and north of Sydney. A major benefit of our approach is that models are fast to run, require simple inputs and Bayesian predictions convey both predicted $PM_{2.5}$ and associated prediction uncertainty. Future research could include the application of similar methods to other regions, collecting more data to improve model precision and developing Bayesian $PM_{2.5}$ models for wildfires.

## Introduction

Airborne particulate matter less than 2.5 μm in diameter ($PM_{2.5}$) is a pollutant small enough to cause cardiovascular and respiratory problems in humans, which can be estimated regionally through increased hospitalisations and premature deaths [1, 2]. $PM_{2.5}$ levels can spike due to smoke from forest fires, both wildfires and hazard reduction burns (HRB), which has been shown to impose substantial costs on healthcare systems [3, 4]. Tools to estimate $PM_{2.5}$ associated with forest fires have a critical role in fire planning, response and issuance of health advice.

are freely available from the New South Wales Government at free online at https://data.airquality.nsw.gov.au/docs/index.html. Information on ERA 5 gridded reanalysis weather product download is at https://www.ecmwf.int/en/forecasts/datasets/reanalysis-datasets/era5.

**Funding:** This research was funded by the NSW Department of Planning, Industry and Environment, via the NSW Bushfire Risk Management Research Hub. Air quality data was accessed from the NSW Department of Planning, Industry and Environment, who manage the NSW air quality monitoring network. The funders had no role in study design, data analysis, decision to publish, or preparation of the manuscript.

**Competing interests:** The authors have declared that no competing interests exist.

HRBs are carried out as a wildfire mitigation measure. The aim is to reduce fuels in order to reduce ignition risk, slow future wildfire spread, reduce wildfire intensity and ultimately protect communities. However, a considerable side-effect of HRBs is that smoke pollution can affect large populated areas. For example, in Sydney, Australia, smoke from HRBs can flow into and sit over large areas of the city and suburbs. In one example, during May 2016 HRBs caused several very smoky days over Sydney, where mean daily $PM_{2.5}$ of all monitors was 24 $\mu gm^{-3}$ and there were an estimated 14 premature deaths and 29 and 50 cardiovascular and respiratory hospitalisations respectively [5]. Given the potential impacts, HRB programs must consider where smoke goes, the level of community exposure to pollutants and ways to reduce smoke risk, and weigh the smoke risks with the wildfire risk reduction benefits of HRBs.

Weather influences the timing, ignition and fire behaviour at an HRB. HRBs need to be scheduled during a limited range of weather conditions that allows fire to spread at a rate and intensity that is containable. Weather also influences smoke production by controlling fire intensity and fuel consumption rates [6, 7], with fuel consumption rates increasing as intensity increases [8]. Weather also determines smoke dispersal patterns: e.g. wind speeds, directions, boundary layer heights and broader synoptic patterns influence vertical and lateral smoke movement [9, 10]. Scheduling an HRB can be a complex decision for a fire agency, given they must consider how weather influences the success of an HRB and smoke dispersal, in addition to other factors such as fuel conditions, crew availability and risk to nearby houses. Tools to better assess the likelihood of negative smoke effects from HRBs under different weather conditions would substantially improve the ability of fire agencies to better understand pollution risks and safely complete HRB programs.

In different parts of the world, fire agencies currently use physics-based smoke dispersion models to predict community exposure to smoke, including $PM_{2.5}$. An advanced example is the BlueSky system in the USA which links fire, emissions and dispersion models to predict regional smoke pollutant concentrations [11]. In NSW, CCAM-TAPM is used to model smoke dispersion and pollutant concentrations from HRBs. It is a combination of an advanced dispersion model TAPM (The Air Pollution Model, [12]) and regional climate model CCAM (Conformal Cubic and Atmospheric Model, [13]). Evaluations of such systems are rare in the published literature, as are studies of $PM_{2.5}$ related to HRBs in general [14]. However, such models have been found to have a limited ability to predict $PM_{2.5}$ levels from forest fires [15, 16]. Physics-based models also tend to require a large amount of computing power, can take a long time to run predictions and require specialists to run the models and interpret the outputs, all of which limit ease-of-use operationally.

An alternative to physics-based models are empirical models, which by comparison can be simple to run and have short computation times. Such models produce predictions of a single variable of interest based on observed data. These can be developed at different scales. However, producing highly accurate empirically based predictions from individual fires may not be realistic due to difficulties in collecting enough smoke-pollution observations and relating them to a particular fire: e.g. sparse monitoring networks (mostly clustered in cities) making detection from any particular fire unlikely and complex weather circulation patterns between a fire and a monitor. Some authors have produced empirical models of $PM_{2.5}$ based on observations of monitors placed near HRBs, but these have been limited to within several kilometres of HRBs [17, 18]. Modelling at the individual-fire scale is complex, but predicting at a coarser regional-scale is likely to be more feasible and produce models applicable to the worst pollution days: i.e. when smoke is affecting a large area rather than just an individual station.

Here, we focused on developing a probabilistic model of daily $PM_{2.5}$ across the Sydney area using Bayesian regression. The aim was to produce a model that provides a general, fast-to-run and operationally useful assessment of daily $PM_{2.5}$ levels in the Sydney basin (Australia) as a

function of regional fire and weather variables. The model is aimed as an operational tool that can complement existing air quality forecast tools to support decisions around the scheduling of HRBs near Sydney. For example, the model could help identify conditions where conducting HRBs requires further consideration of smoke effects before proceeding, which may be assessed in light of predictions from existing models (e.g. physics-based models). We used Bayesian modelling to produce probabilistic predictions that provide highly informative model outputs and provide a clear indication of prediction uncertainty.

## Methods

### Study area and fire data

We focused on the area around Sydney in New South Wales, Australia. Sydney has > 4 million residents and is situated in a low-lying basin between the coast and large tracts of eucalypt forest, including the Greater Blue Mountains World Heritage Area [19]. A combination of conducive geographic and weather characteristics make Sydney subject to long-lasting pollution events [20], with the worst events usually stemming from forest fires [21]. The forests around Sydney are wildfire-prone: many wildfires have burnt thousands of hectares and destroyed hundreds of homes, such as in the years 2002–03, 2013 and 2019–20 [22, 23].

We analysed fire activity within a 150 km buffer of the Chullora (a suburb of Sydney) air quality monitoring station (AQS). Chullora AQS is centrally located among five AQS in the area with longer-term $PM_{2.5}$ records matching our study period (see below) and 150 km captures most of the Blue Mountains World Heritage Area (Fig 1). Our study period was 2012 to 2021 because this was the period for which VIIRS SNPP (Visible and Infrared Imager/Radiometer Suite, Suomi) satellite hotspots, our measure of fire activity, was available. As our interest was in smoke related to HRBs, our analysis was limited to March to September when most HRBs are usually conducted.

For all March-September days from 2012 to 2021, we first identified days with active fires in the study area using VIIRS SNPP hotspots [24]. Active fire days were those with at least one cluster of three or more hotspots, with hotspots grouped into a cluster when within 5 km of each other (S2 Appendix). We used hotspot clusters to identify active days so as to exclude days where only one or two isolated hotspots were recorded, which may have been from very small non-HRBs such as wood heaps burning on farmland. We used VIIRS instead of MODIS hotspots because VIIRS is more fire-sensitive and has higher resolution (375 m vs. 1km at nadir), which allows more consistent detection of active fires [24, 25]. We did not use fire-agency fire-history data because it does not include a record of daily fire activity, only final burnt area, fire ignition date and fire containment date.

We generated fire area variables based on VIIRS hotspots as predictors for our statistical modelling. VIIRS captures hotspots during the day (~1 pm to ~3 pm) and night (~12 am to ~2 am). Often VIIRS captures hotspots once per location during the day and once at night, but two VIIRS swathes can overlap at the edges, meaning hotspots for one location are captured twice during the day and/or night. For each active fire day identified, we calculated the daily total hotspot (day + night) area inside the Chullora 150 km buffer which, due to potentially overlapping hotspots, was derived by conducting a GIS intersect of all hotspots for a 24-hour period with a 500 m x 500 m grid (25 ha) covering the study area. The total daily fire area was the total number of unique intersecting grid cells * 25 ha. Note that we counted night hotspots captured in the early morning (before 7 am) toward the prior day's fire area. For statistical modelling predictors, we split fire area into sectors: fire area west (hotspots that were 270° ± 45° from Chullora), fire area north (360° ± 45°) and fire area south (180 ° ± 45°) (Fig 1). We chose to include area using this method after we also explored simply using total fire area in

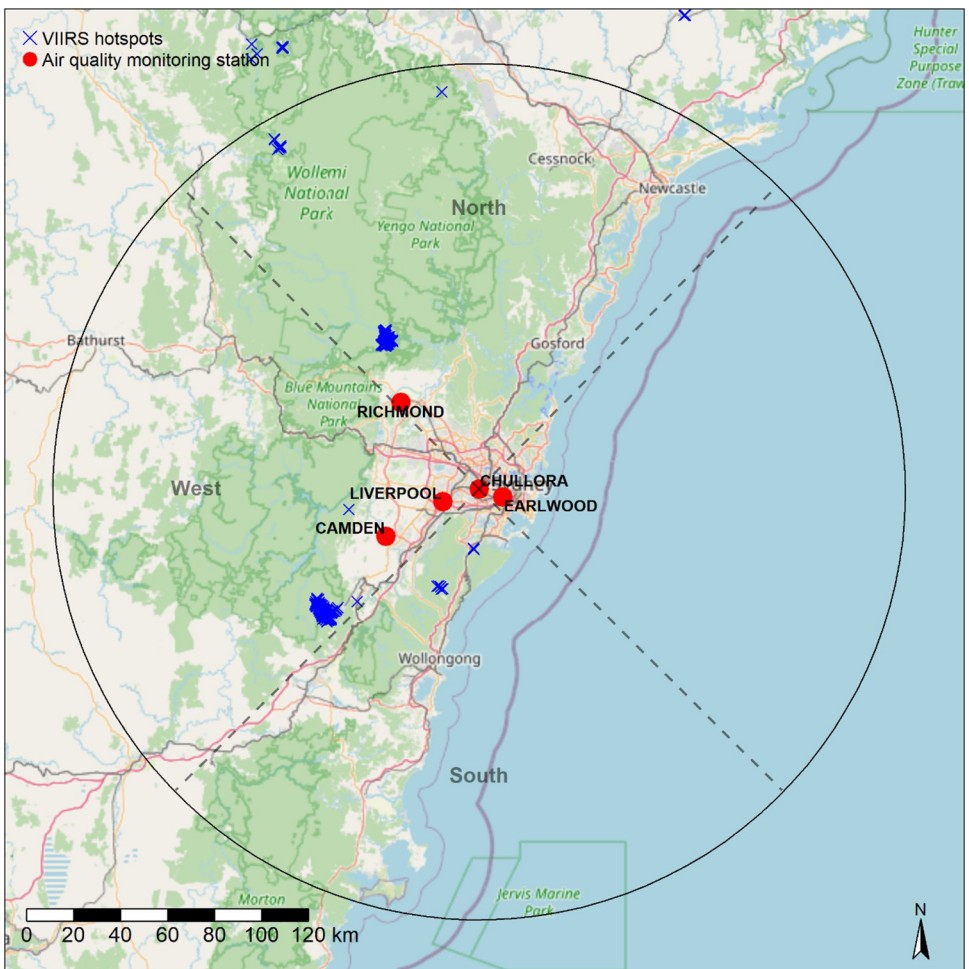

**Fig 1. Study area map showing air quality monitoring stations used in analysis and example of VIIRS SNPP hotspots for one day and night between 2019-04-28 midday and 2019-04-29 midday.** 150 km buffer around Chullora is black circle. West, South and North region were used for calculating separate fire area variables. Basemap: © OpenStreetMap contributors.

the region, but found splitting fire area by direction produced better models. A limitation that should be noted is that hotspots may not be detected for fires burning under heavy cloud. This means very cloudy days with fires may have been wrongly determined as having no fire (thus not included in modelling), or fire in parts of the study area may have been missed.

## PM$_{2.5}$ and weather data

We downloaded all available PM$_{2.5}$ data measured at air quality monitoring stations (AQS) from the NSW Department of Planning, Industry and Environment for the period 2012–2021, freely available at https://data.airquality.nsw.gov.au/docs/index.html. PM$_{2.5}$ data is available before 2012, but we did not use this because VIIRS hotspots are only available from 2012. From the data, we determined that there were five AQS within the Sydney area that had records from 2012 to 2021: Chullora, Liverpool, Richmond, Camden, and Earlwood (Fig 1). The AQS data contained hourly averages of PM$_{2.5}$, among other measurements that were not used here including ozone, PM$_{10}$ and weather recorded at the AQS.

We calculated mean daily $PM_{2.5}$ for each of the five AQS. We used daily means because there are reportable standards that currently exist for daily mean $PM_{2.5}$. The Australian National Air Quality Standards state that mean daily $PM_{2.5} > 25$ μgm$^{-3}$ at a single AQS is considered a reportable exceedance [26]. For our analysis, a daily period was from midday to midday to align with the cycle of fire activity and smoke output: in general, afternoon HRB ignition and smoke effects from the afternoon until the next morning.

In addition to VIIRS-based estimates of fire area, we sampled several variables as predictors for the modelling (Table 1). Weather was from ERA5, which is a gridded (30 km) hourly atmospheric reanalysis product with numerous weather variables available for surface and atmospheric levels [27, 28]. We chose variables that are used in fire behaviour and air quality prediction: U and V wind components (from which wind speed and direction are derived), temperature, planetary boundary layer height, total cloud cover and mean sea level pressure over the Tasman Sea and western Sydney [29, 30]. We sampled ERA5 variables from an ERA5 pixel located over western Sydney (Penrith). For the wind U and V components, we sampled from two additional locations to capture potentially differing wind effects along the coast and inland: 1) an ERA5 pixel on the Sydney coast (centroid near Sydney International Airport) to capture sea breezes, and 2) an ERA5 pixel inland in the Blue Mountains over Katoomba. A negative U component of wind ("U-wind") indicates wind from the east (+ from west) and a negative V component ("V-wind") indicates wind from the north (+ from south). We calculated wind speed for modelling from the wind component variables sampled over western Sydney only. However, during initial analysis we found a strong correlation between wind speed and planetary boundary layer height, so instead we used the product of wind speed and planetary boundary layer height, known as ventilation index [31].

We calculated daily mean values for all variables used in modelling, except for the coastal U-wind and V-wind where we calculated mean values for the afternoon (2 pm to 6 pm local time) to capture afternoon sea breeze occurrence. We initially tested using mean values for sub-daily periods (afternoon, night and morning) for all weather variables but found that: a)

**Table 1. Predictor variables used in modelling.**

| Variable | Description | Unit |
|---|---|---|
| **Lag PM$_{2.5}$ (Daily)** | A lag of the mean Sydney $PM_{2.5}$ value, i.e. mean $PM_{2.5}$ from the prior 24 hrs. This was used for both the mean and maximum Sydney $PM_{2.5}$ models. | μgm$^{-3}$ |
| **Ventilation index (Daily)** | Product of wind speed (m s$^{-1}$) and planetary boundary layer height (m) from ERA5 cell over western Sydney (lon: 150.7, lat: -33.76). | m$^2$ s$^{-1}$ |
| **Temperature (Daily)** | Mean daily temperature from ERA5 cell over western Sydney. | Celsius |
| **Total cloud cover (Daily)** | Total cloud cover from ERA5 cell over western Sydney, as proportion 0–1. | proportion |
| **Mean Sea Level Pressure (Daily)** | Mean daily value of mean sea level pressure (MSLP) sampled in two locations for two separate variables; 1) western Sydney, 2) middle of Tasman Sea, mid-way between Sydney and New Zealand (lon: 161.71, lat: -36.04). | hectopascal |
| **Coastal wind components U and V (Afternoon, 2 pm to 6 pm)** | Wind component variables sampled from ERA5 grid cell on Sydney coast (lon: 151.3, lat: -33.9) to detect presence of an afternoon sea breeze. | wind vectors |
| **Inland wind components U and V (Daily)** | Wind component variables sampled from ERA5 grid cell in the Blue Mountains west of Sydney, over Katoomba (lon: 150.31, lat: 33.71). | wind vectors |
| **Fire area west, south and north (Daily)** | Three separate variables. From VIIRS hotspots within 150 km, but in each direction from Chullora. West = 270$^o$ ±45$^o$ from Chullora, south = 180$^o$ ±45$^o$, north = 360$^o$ ±45$^o$. | hectare |

there was generally a high correlation between the values for different periods, and b) using daily values produced similar performing models to those with the sub-daily predictors. Thus, we used daily means for all variables, except for the afternoon sea breeze variables (U and V coastal wind). We included a lagged $PM_{2.5}$ term in the models to account for temporal auto-correlation, i.e. so that the model considered the influence of the previous day's $PM_{2.5}$ on the subsequent day's $PM_{2.5}$. Daily $PM_{2.5}$ is likely to be temporally auto-correlated as pollution can build up over several days over Sydney before it clears away [32].

## Modelling approach

There were a total of 597 rows for active fire days between March and September in the years 2012–2021. 80% (477 days) were used for model training and 20% (120 days) for model testing. We derived one model of the mean of the daily (i.e. midday to midday) mean $PM_{2.5}$ values from the five AQS ("mean model"), which indicates raised $PM_{2.5}$ across the whole of the Sydney area (Fig 1). We derived a second model for the maximum from the five AQS daily mean $PM_{2.5}$ values ("maximum model"), which indicates a high daily $PM_{2.5}$ at any one of the five AQS. Note for model one, the mean of all AQS could be skewed by an extremely high value at one or two AQS, thus a high Sydney-wide mean does not necessarily affect all AQS. However, this approach would still provide a useful indication of more widespread pollution.

Our approach to fitting the models had two stages that involved fitting Generalised Additive Models (GAMs) followed by fitting Bayesian models. A GAM is a form of regression model that uses smoothing splines instead of simple linear effects [33]. Smoothing splines allow non-linear effects to be incorporated into a model, which has been found to be a good approach for air pollution research due to common complex non-linearity in the data [34, 35]. Bayesian models produce predictions in the form of a distribution of plausible values for a given set of predictor conditions based on the model training data, rather than a point estimate as in deterministic models. This provides flexibility in presenting and interpreting the predictions, as the predictive distribution can be summarised in a range of ways depending on user need (e.g. median, "credible intervals", "highest posterior density interval", chance that outcome will be $> x$) [36]. Our end goal was to fit Bayesian models using the "brms" package in R with smoothing splines for the predictors [37].

We initially tried running a model selection process in the "brms" package but found that fitting all models with smooths for all predictors was too time-consuming to be practical. Instead, we used the "mgcv" package in R [38] to fit GAMs (generalised additive models; Gamma family with log link) for a model selection process. We then fitted the best GAMs as Bayesian models using the "brms" package, i.e. the model formula from the best GAM was used to fit a Bayesian model via "brms". The "brms" package uses the smooth functions from "mgcv" in model fitting.

For the GAM model selection process, we fitted all combinations of predictors as separate models and retained the Akaike Information Criterion (AIC) for each model [39]. The U-wind and V-wind variables were tested in this process as interaction terms, i.e. two-dimensional smooths with the "mgcv" syntax of "s(U coastal wind, V coastal wind)" and "s(U Katoomba wind, V Katoomba wind)". We did this because both U and V components are required to understand wind speed and direction, including when making predictions.

We judged the best model to be that with the lowest AIC. To ensure there was no substantial drop in predictive accuracy from the model selection process, we compared the correlation-based coefficient of determination (i.e. $R^2$) from test set predictions with the best model to $R^2$ from test set predictions with the full model (all predictors included). Finally, we fitted the best model as a Bayesian model using the "brms" package in R. Bayesian models were fitted

with four chains for 10000 iterations per chain, thinning of 10 and discarded warmup of 5000. We ensured model convergence using the Gelman-Rubin convergence diagnostic (all Rhat = 1) and inspection of MCMC plots.

In the results, we have reported on the GAM (deterministic) prediction accuracy of the best models and the models with all predictors included. We have provided plots of model effects and example probabilistic predictions from the Bayesian models.

## Results

### Descriptive analysis

Mean Sydney $PM_{2.5}$ was situated mostly between 5 $\mu gm^{-3}$ and 15 $\mu gm^{-3}$ (mean 10 $\mu gm^{-3}$), with a maximum of 47 $\mu gm^{-3}$ and 90th percentile of 16 $\mu gm^{-3}$ (Fig 2). Only 3% of days were > 25 $\mu gm^{-3}$. Maximum Sydney $PM_{2.5}$ was clustered between 5 $\mu gm^{-3}$ and 25 $\mu gm^{-3}$, with a maximum of 142 $\mu gm^{-3}$ and 90th percentile of 24 $\mu gm^{-3}$. There were 8% of days > 25 $\mu gm^{-3}$ and 12 days > 50 $\mu gm^{-3}$.

In initial analysis of individual predictors against mean Sydney $PM_{2.5}$, some predictors appeared to have strong effects (Fig 3). Maximum Sydney $PM_{2.5}$ had similar relationships with individual predictions, so we focus only on mean Sydney $PM_{2.5}$ in this section. High mean $PM_{2.5}$ only occurred when ventilation index was low. When daily temperature was > ~17 C, mean $PM_{2.5}$ was generally low. Higher mean $PM_{2.5}$ mostly occurred when coastal afternoon U-wind was negative, i.e. when an afternoon sea breeze was occurring. A positive U-wind at Katoomba indicates more westerly winds flowing towards Sydney. High mean $PM_{2.5}$ occurred mostly when U-wind at Katoomba was positive, indicating winds were more from the west than from the east during high $PM_{2.5}$ days. Lagged $PM_{2.5}$ had a strong association with mean $PM_{2.5}$. For the three fire area variables, there were positive associations with $PM_{2.5}$ for all in the 0 to 3000 ha range, where most data was situated, and then differing effects at larger areas,

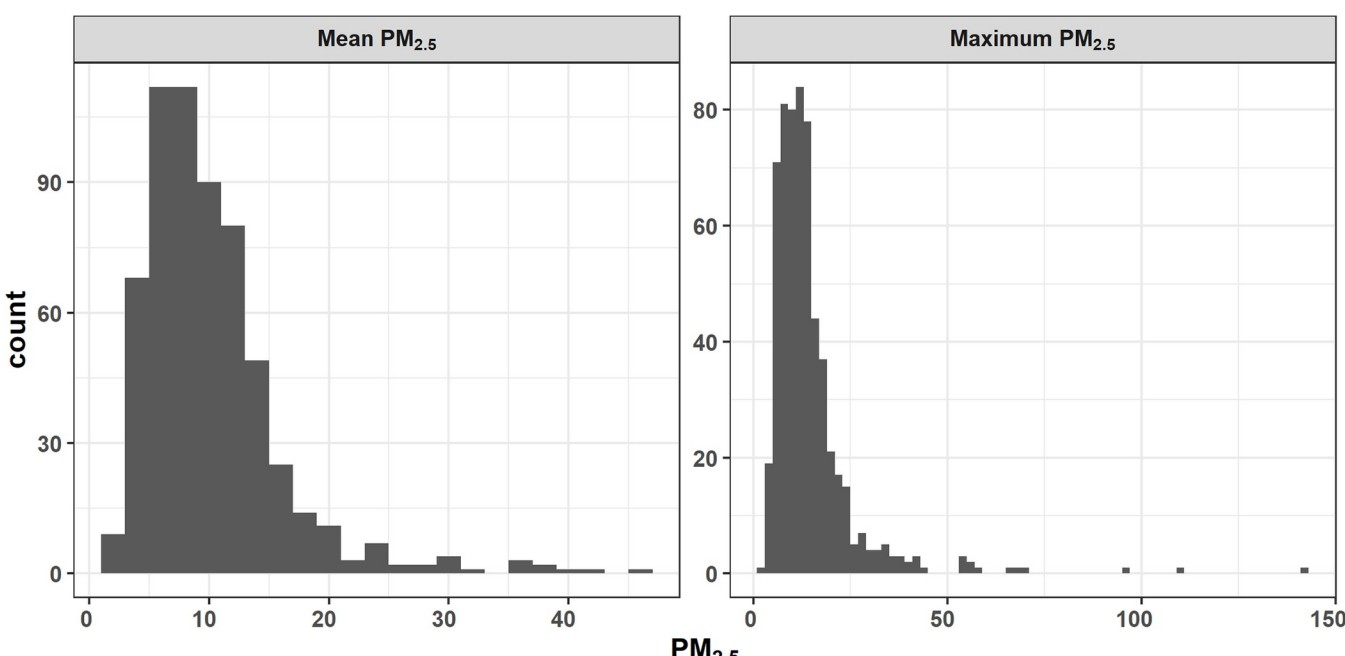

**Fig 2. Histograms of daily mean $PM_{2.5}$ of all five AQS and maximum daily $PM_{2.5}$ from all five AQS (i.e. value from AQS with highest daily mean $PM_{2.5}$).** "Daily" here means midday to midday.

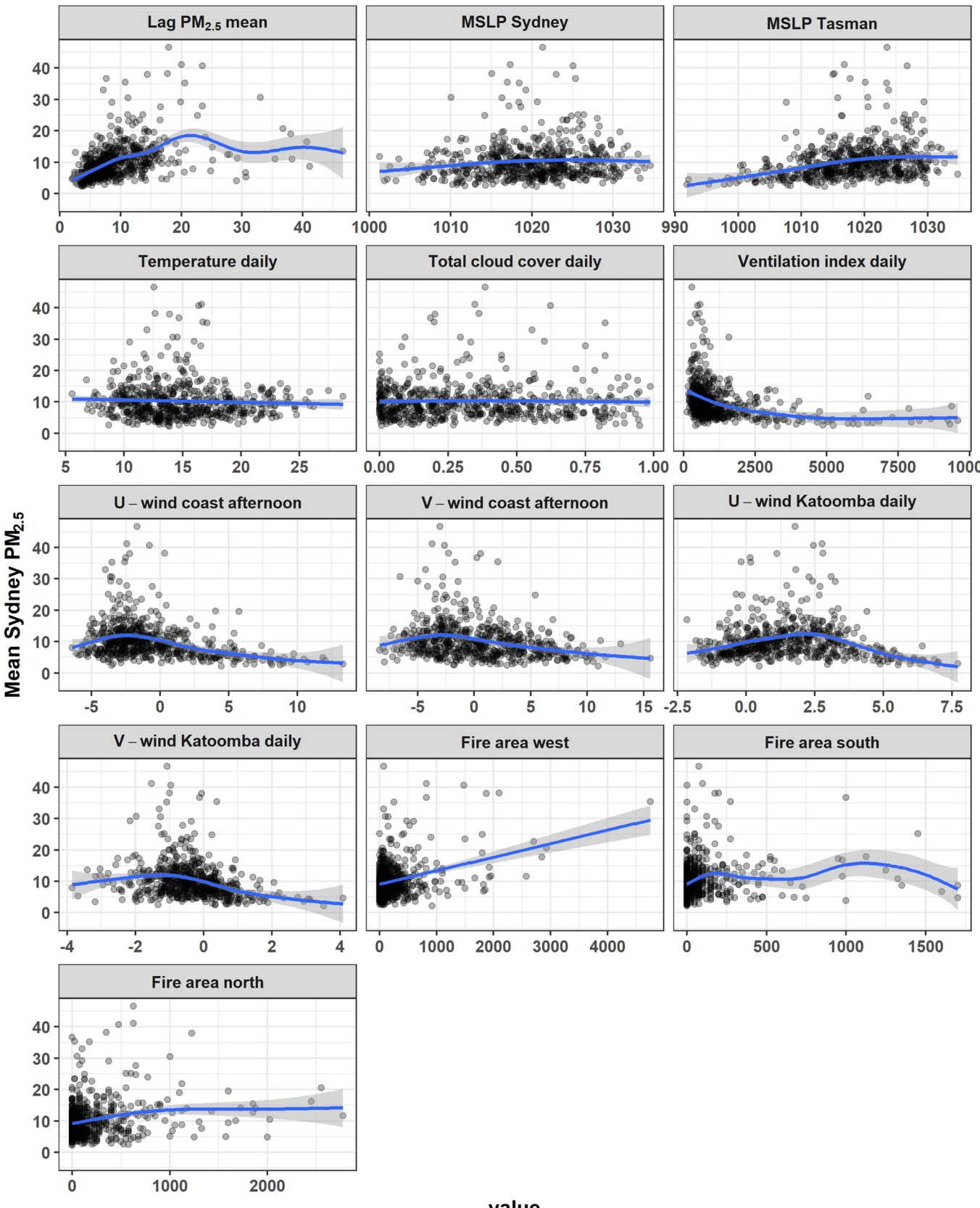

**Fig 3. Scatter plots of mean Sydney PM$_{2.5}$ vs predictor variables, with single variable GAM smooths fit via the "mgcv" package in R.**

**Table 2. GAM accuracy results for models with all variables and best models from the model selection process.**

| Dependant | Model | AIC | Deviance Explained | $R^2$ | $R^2$ test set |
|---|---|---|---|---|---|
| Sydney Mean | All variables | 2379.5 | 67.9% | 0.580 | 0.580 |
| | Best | 2376.0 | 67.9% | 0.580 | 0.564 |
| Sydney Max. | All variables | 2879.5 | 63.7% | 0.516 | 0.444 |
| | Best | 2873.5 | 63.7% | 0.516 | 0.443 |

although only a few observations were > 3000 ha. Although there were generally positive associations, there were still many days with low fire area that had high $PM_{2.5}$.

## Model selection and predictor effects

For both mean and maximum Sydney $PM_{2.5}$, the best models from the GAM-based selection contained all predictors except the MSLP variables and total cloud cover. The selected models were superior to the full models in terms of AIC and their predictive power was only marginally different (Table 2).

When fit as Bayesian models, the predictors for both the mean and maximum models had generally similar shapes and directions of effects. Both are discussed here but plots for the mean $PM_{2.5}$ model are included in the main text and plots for the maximum $PM_{2.5}$ model are in S1A Fig in S1 Appendix. Lag mean $PM_{2.5}$ had a strong positive effect between 0 $\mu gm^{-3}$ and 20 $\mu gm^{-3}$ in both models. Above a lag of 20 $\mu gm^{-3}$, the effect became more uncertain as indicated by wider bounds (Fig 4A). In the mean model the effect flattened (Fig 4A) and in the maximum model the effect was weaker but still positive (S1Aa Fig in S1 Appendix). In both models, fire area variables had strong positive effects, although to the south and north > ~1500 ha there was no effect or a negative effect with low confidence (wide bounds) (Fig 4D-4F, S1Ad-S1Af Fig in S1 Appendix), which reflected the low number of days with such large fire areas. In both models, the afternoon coastal wind variables (U and V), when transformed to wind direction and speed, indicated that winds from the east (i.e. a sea breeze) increased $PM_{2.5}$ and the effect increased with wind speed (Fig 4G, S1Ag Fig in S1 Appendix). The daily inland (Katoomba) wind variables indicated that the highest $PM_{2.5}$ levels were associated with north-westerly winds, with the effect increasing with wind speed in both models (Fig 4H, S1Ah Fig in S1 Appendix). Easterly winds inland meant lower $PM_{2.5}$, which decreased as wind speed increased. While the highest $PM_{2.5}$ was associated with the lowest daily mean temperature for both models, temperature had a curved effect (Fig 4C, S1Ac Fig in S1 Appendix). The lowest $PM_{2.5}$ was at ~19 C, with increases in $PM_{2.5}$ associated with both lower temperatures (stronger effect) and higher temperatures. In both models, a low ventilation index over western Sydney was associated with high $PM_{2.5}$, particularly < ~2000 $m^2\ s^{-1}$. Above ~2000 $m^2\ s^{-1}$, the effect levelled out and had very wide credible interval bounds (Fig 4B, S1Ab Fig in S1 Appendix), indicating a large deal of uncertainty due to a low number of observations at this level in the model data (Fig 3).

## Model predictions

Plots that demonstrate practical applications of the models are shown below. Predictions are shown as a) plots of the entire predictive distribution when levels of a single variable are varied in Fig 5, and b) a gridded or tiled output where the levels of multiple variables are varied, but predictive distributions are summarised based on the chance of $PM_{2.5}$ exceeding a threshold, i.e. percent of predictive distribution greater than a threshold (Fig 6, S1B Fig in S1 Appendix).

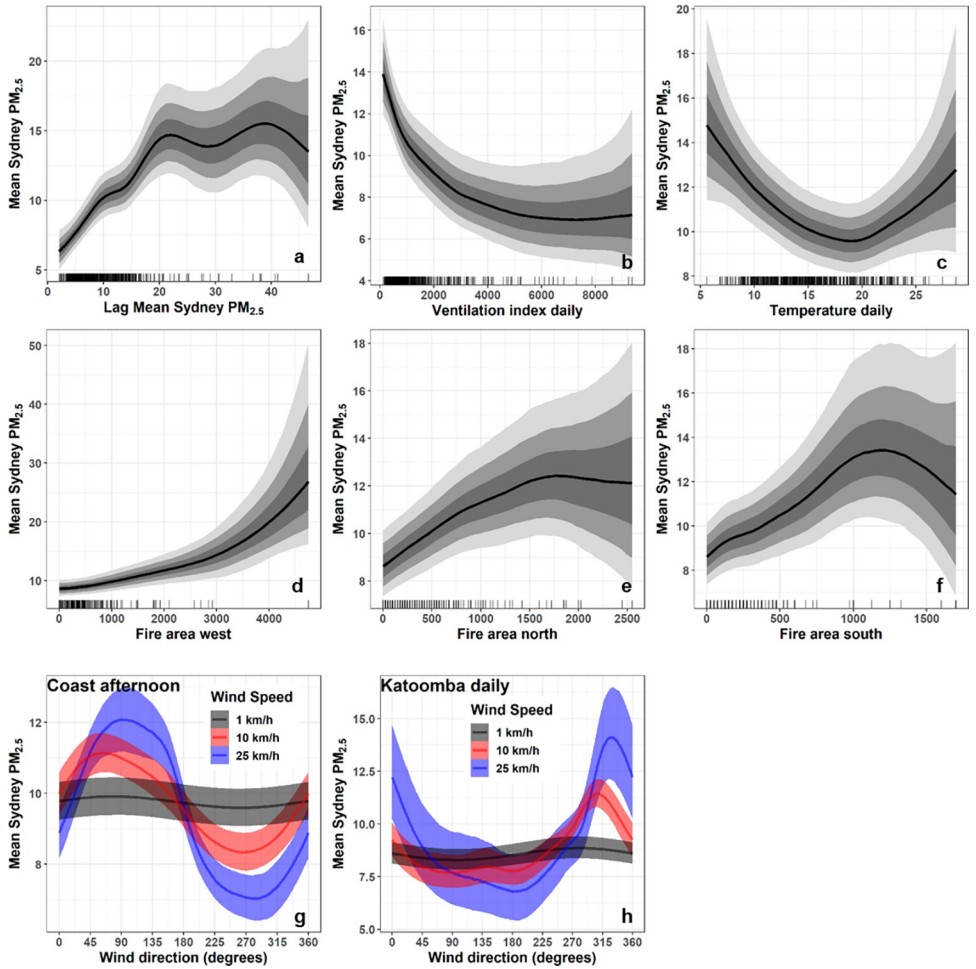

**Fig 4. Bayesian mean effects plots for each variable for mean daily Sydney PM$_{2.5}$ model.** The grey bands represent, from darkest to lightest grey, 0.5, 0.8 and 0.95 credible intervals. Effect of each fire area variable is shown with other two fire area variables held at zero. Other variables held at mean values. U and V wind components have been converted to wind speed and direction effects (bands show 0.5 credible interval). Wind directions are standard wind direction in degrees, i.e. 180 = southerly wind.

Fig 5 demonstrates that as fire area in the west increases, mean and maximum Sydney PM$_{2.5}$ are predicted to increase. For both models, the width of the predictive distribution also becomes wider as fire area increases, which reflects increasing uncertainty associated with the predictions. The example shows that under the specified set of conditions (Fig 5), when area in the west increases from 100 ha to 4000 ha, the chance of a mean Sydney PM$_{2.5}$ exceeding 25 μgm$^{-3}$ grows from 2% to 76%, and the chance of mean PM$_{2.5}$ exceeding 15 μgm$^{-3}$ grows from 42% to 98%. There is generally a much greater chance that maximum Sydney PM$_{2.5}$ will exceed the thresholds under the conditions in Fig 5: a 31% to 100% chance of exceeding 25 μgm$^{-3}$, and a 78% to 100% chance of exceeding 15 μgm$^{-3}$. This reflects that it is more likely for one AQS within Sydney to record an exceedance (i.e. the maximum PM$_{2.5}$ model) than it is for the mean of all AQS to exceed the same threshold (i.e. the mean PM$_{2.5}$ model). Note that such plots could be reproduced with any exceedance threshold specified or with any variables set at different levels.

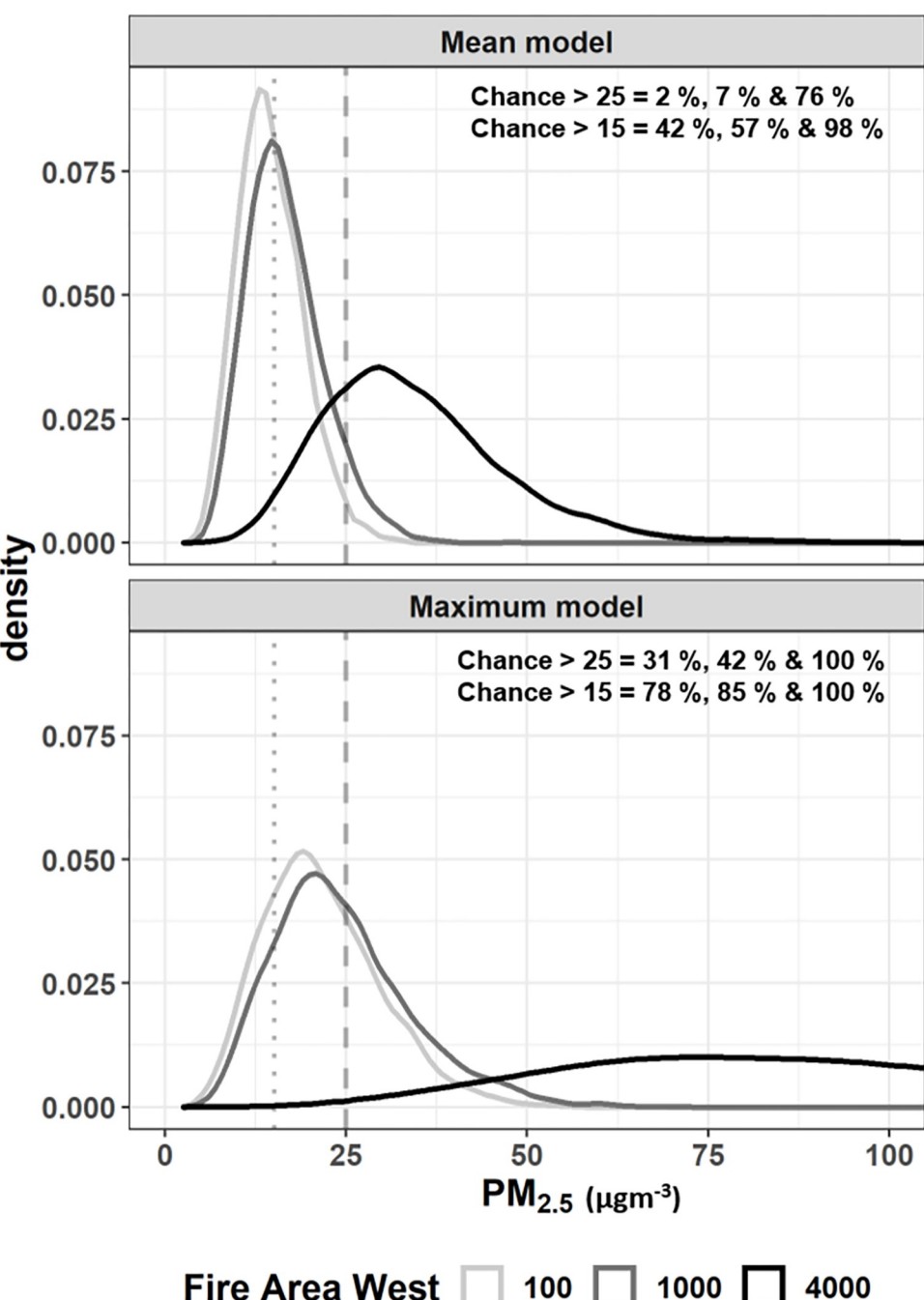

**Fig 5. Predictive distributions from the mean and maximum daily Sydney PM$_{2.5}$ models.** Predictions for three levels of fire area west, with other variables at set levels: Lag PM$_{2.5}$ = 10 μgm$^{-3}$, ventilation index = 500 m$^2$ s$^{-1}$, temperature = 10 C, fire areas south and north = 100 ha, coast wind speed and direction = 10 km h$^{-1}$ and easterly (sea breeze), inland wind speed and direction = 10 km h$^{-1}$ and westerly. Dotted and dashed vertical lines are thresholds defined at 15 μgm$^{-3}$ and 25 μgm$^{-3}$ respectively, with percent of distribution > thresholds indicated in text within the plots.

Amongst the columns in Fig 6A (i.e. levels of fire area west), the predicted chance of mean Sydney PM$_{2.5}$ > 25 μgm$^{-3}$ is generally higher on the right, indicating that increasing fire area increases the chance of a threshold exceedance. Amongst the rows, mean PM$_{2.5}$ is predicted to

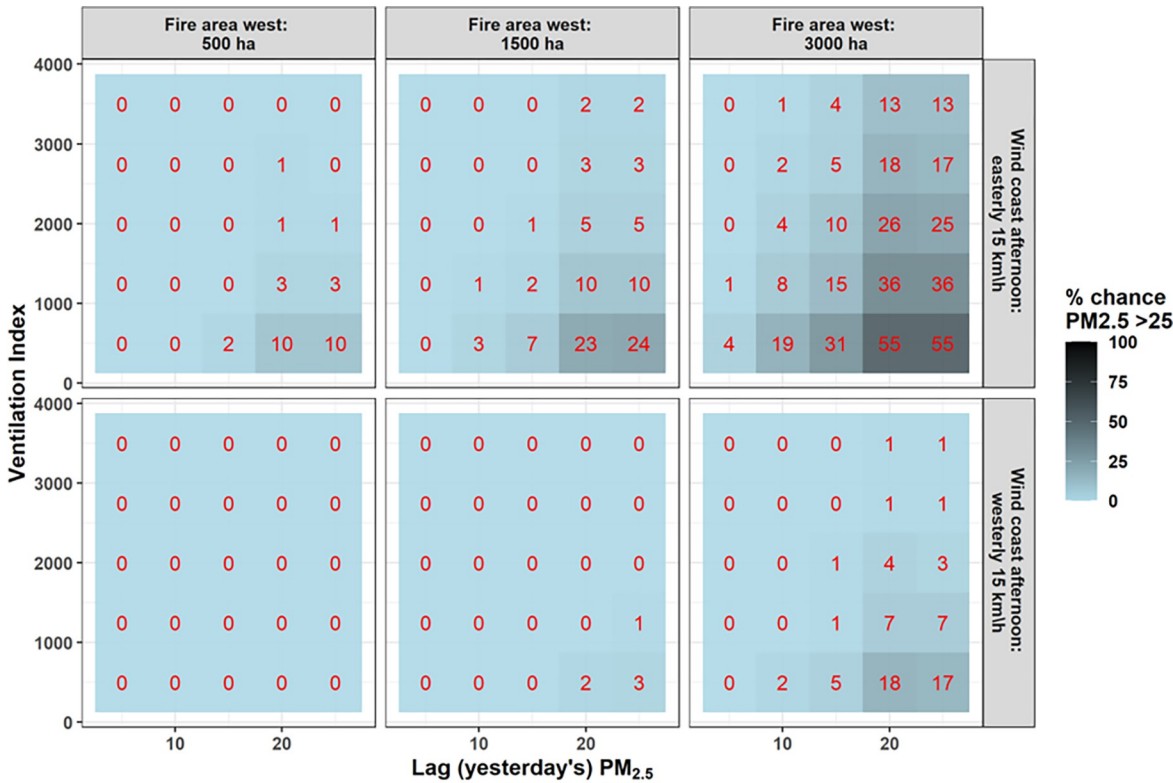

a) Mean PM$_{2.5}$ model predictions

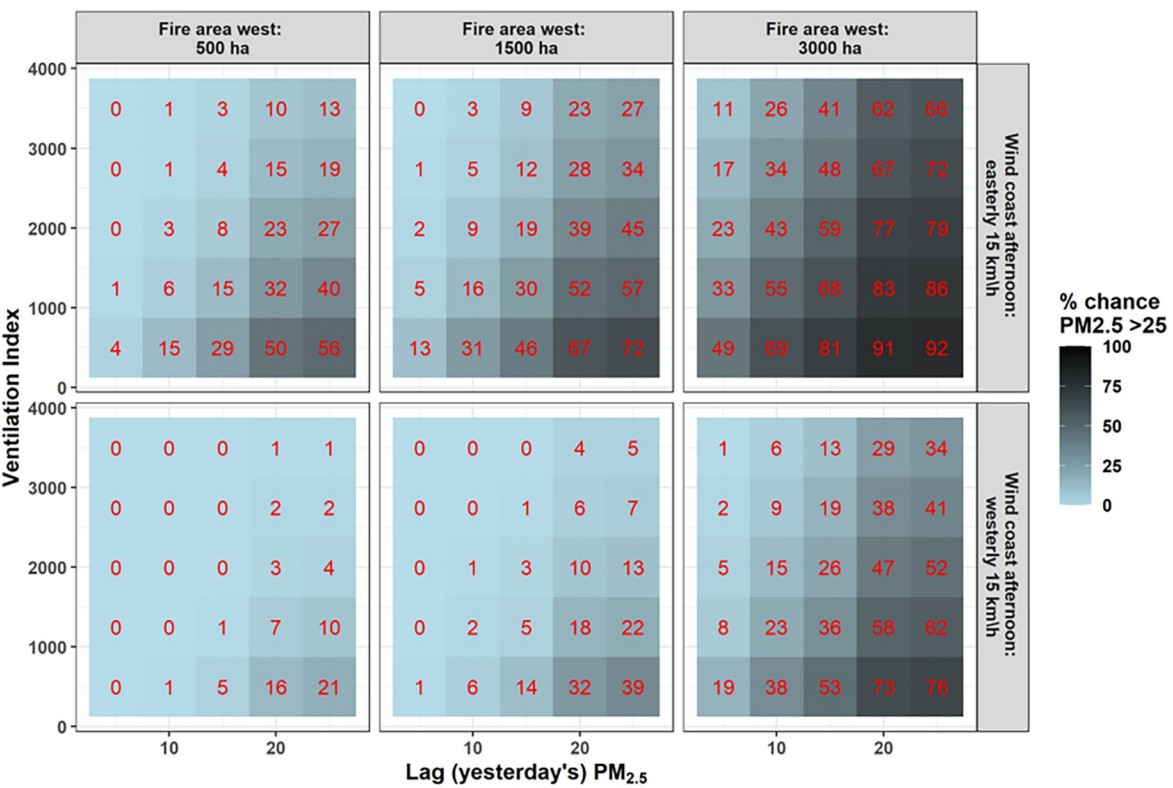

b) Maximum PM$_{2.5}$ model predictions

**Fig 6. Tile plot of predictions for mean Sydney PM$_{2.5}$ model (top) and maximum Sydney PM$_{2.5}$ model (bottom).** Each coloured grid square shows the percent chance, under different predictor conditions, of exceedance above 25 µgm$^{-3}$ (darker = higher chance, also in red text), i.e. percent of predictive distribution > 25 µgm$^{-3}$. Other variables held at: daily temperature 14 C, fire areas north and south = 0 ha, inland (Katoomba) wind speed and direction = 15 km h$^{-1}$ and westerly. Wind speeds and directions were calculated from the relevant U and V wind variables.

be higher when a sea breeze (easterly) is occurring (top row). Of all six panels in Fig 6A, the highest chance of exceeding 25 µgm$^{-3}$ is in the top right, with 3000 ha fire area in the west and a sea breeze. In this panel, there is predicted to be > 50% chance of mean PM$_{2.5}$ > 25 µgm$^{-3}$ when lag PM$_{2.5}$ is high and when ventilation index is low. In each of the panels, the highest chance of exceedance is also the bottom right, with high lag PM$_{2.5}$ and low ventilation index. When fire area in the west is low and there is no sea breeze (Fig 6A, bottom left), there is almost no predicted chance of mean Sydney PM$_{2.5}$ > 25 µgm$^{-3}$. Note that *mean* PM$_{2.5}$ > 25 µgm$^{-3}$ is rare in the observed data (Fig 2), so a plot of the chance of mean PM$_{2.5}$ exceeding 15 µgm$^{-3}$ is included in S1 Appendix for comparison.

The chance of maximum Sydney PM$_{2.5}$ exceeding 25 µgm$^{-3}$ is generally much higher, as this would require only one AQS to have a daily mean exceeding 25 µgm$^{-3}$. However, the pattern is similar in that the greatest chance of a maximum PM$_{2.5}$ exceedance is in the top right panel of Fig 6B (fire area west = 3000 ha, afternoon sea breeze occurring), particularly when ventilation index is low and lag PM$_{2.5}$ is high. Under these conditions there is ~90% chance of an exceedance. There are also significant chances of exceedances in other conditions, with the maximums always being when ventilation index is low and lag PM$_{2.5}$ is high: e.g. ~70% chance at 1500 ha fire area in the west and a sea breeze (easterlies) (top centre, Fig 6B).

## Discussion

We developed and demonstrated Bayesian probabilistic models of mean and maximum daily (midday to midday) PM$_{2.5}$ across Sydney based on fire and weather variables. We expect the models to be useful in the context of scheduling hazard reduction burns in the Sydney area, particularly identifying if a day has a high chance of exceeding hazardous PM$_{2.5}$ thresholds. This would indicate that an HRB would require further consideration before ignition or the HRB should be postponed. Our use of Bayesian modelling means that predictions can be presented in a range of formats depending on user need: e.g. plots of predictive distributions (Fig 5) or calculating the chance of a PM$_{2.5}$ threshold exceedance (Fig 6). The models were developed with March to September data, so are suitable for the main HRB period but not for summer wildfire-related predictions.

The lag PM$_{2.5}$ variable (prior 24 hr mean PM$_{2.5}$ for both models) had a strong effect on PM$_{2.5}$ levels in both models. This suggests that smoke can build up over multiple days around Sydney, although the effect flattened out in the mean PM$_{2.5}$ model above lag PM$_{2.5}$ of 20 µgm$^{-3}$. Conducting multiple days in a row of large HRBs, either multiple HRBs or one HRB split into sections may therefore lead to higher PM$_{2.5}$ levels in Sydney. Further research would be required to test the effect of multi-day burning on PM$_{2.5}$ levels.

We found that location of the HRBs was important. Fire area in the west had a strong positive effect, whereas north and south fire area had positive effects in the 0–1500 ha range, above which there was a great deal of uncertainty due to a low number of observations. Differences here were expected because the best conditions for HRBs in the Sydney area are with light west to northwesterly winds, which would push smoke from HRBs in the west or north-west over Sydney. An HRB in the south under westerly or northwesterly winds would carry smoke over the ocean away from Sydney. However, there are cases of multi-day burns and wildfires in the south affecting Sydney's air quality.

The models are location specific to the Sydney area, but some of the effects found are likely generalizable and have been described in other studies. We confirmed that the ventilation index, which has been used as an indicator of air quality potential around fires in the USA [31], is useful in the context of $PM_{2.5}$ modelling. A low value of ventilation index describes conditions of low wind speed and low boundary layer, which have been related to higher pollution levels in other studies [40, 41]. Di Virgilio (2018) also found low PBLH was a consistently important predictor of $PM_{2.5}$ in relation to HRBs in the Sydney area. However, Duc, Rahman [42] found weak correlations between PBLH and $PM_{2.5}$ for a short period in their study, which indicates that other factors are important. We also found that a lower daily temperature was indicative of worse $PM_{2.5}$, which was also found in a study of $PM_{2.5}$ near several HRBs in the Sydney area [18].

Jiang, Scorgie [9] described that the interaction between synoptic scale weather and regional weather determines local air quality in Sydney. We did not find a strong link between $PM_{2.5}$ and MSLP at Sydney and over the Tasman, whereas other authors have linked high-pressure east of Sydney/over the Tasman to higher pollution in Sydney [10, 30]. It may be that by directly using local weather in our modelling, we accounted for any influence of MSLP and MSLP is a surrogate for local weather. We found a sea breeze to be an important determinant of $PM_{2.5}$. There is support for this because several studies have suggested sea breezes can elevate pollution levels over coastal cities, either by trapping or recirculating pollutants [10, 40, 43]. We also found that a daily trend of westerly to north-westerly winds in the Blue Mountains west of Sydney was associated with higher $PM_{2.5}$. Higher $PM_{2.5}$ can therefore be expected when westerlies inland are pushing smoke towards Sydney and a sea breeze traps or recirculates smoke. The effect is likely to depend on how much and how long an HRB burns through the night [18]. In future studies, the length of night burning may be possible to determine with high temporal resolution satellite data, such as Himawari 8 which captures every 10 minutes [44]. Sea breezes may not have the same impact at more intense fires where high smoke plumes carry smoke above the surface level. It should be noted that there is some association between our predictors involving wind speed in our data, mainly that ventilation index (sampled over western Sydney) is correlated with daily wind speeds over Katoomba (calculated from U and V, $R = 0.71$). This means that when ventilation index is low, wind speed at Katoomba is also generally low. Our models are therefore likely to predict poorly for unusual conditions, e.g. ventilation index low but very high wind speeds at Katoomba. We expect this would not be an issue for HRB-related prediction, given that HRBs are only conducted on lower wind speed days.

Our use of Bayesian modelling has operational-use advantages: short compute times (a few seconds), they only require easily available/simple forecast weather variables to run predictions (wind, temperature and boundary layer height for ventilation index, plus yesterday's mean $PM_{2.5}$), and there is flexibility in the way predictions can be presented and communicated. This is advantageous over physics-based simulations that require high-performance computers, take hours to run and require expert interpretation. The models presented here could be used alongside existing methods to create multiple lines of evidence for fire planners to use when scheduling HRBs.

We have demonstrated two alternatives for presenting predictions (Figs 5 and 6), but the best way to communicate predictions would need to be operationally assessed. The particular model that is most useful would also depend on user need, with the best approach probably to consider both the minimum and maximum Sydney $PM_{2.5}$ models in decision-making, as well as predictions via other models. The mean Sydney $PM_{2.5}$ model would give a better picture of potential pollution affecting all or most of the Sydney area, although large outliers can also result in a high mean. The maximum Sydney $PM_{2.5}$ is for understanding where at least one of

the AQS may have high pollution levels. For what constitutes concerning $PM_{2.5}$ levels, some consideration is required. The national standard for an exceedance is for daily mean $PM_{2.5} >$ 25 $\mu gm^{-3}$ at a single AQS [26], which is a suitable threshold for presenting predictions for our maximum $PM_{2.5}$ model (e.g. Fig 6B). However, the mean $PM_{2.5}$ model is for the daily mean of multiple AQS, which is much more rarely $> 25$ $\mu gm^{-3}$: 3% of mean Sydney $PM_{2.5}$ was $> 25$ $\mu gm^{-3}$, but 8% of maximum Sydney $PM_{2.5}$ was $> 25$ $\mu gm^{-3}$ (Fig 2). This simply means that a lower threshold may also be useful when summarising mean $PM_{2.5}$ model predictive distributions (Fig 6) to identify potentially hazardous days for HRBs, but this would be up to a model-user to determine (see examples in S1B Fig in S1 Appendix).

## Conclusion

We have demonstrated two models that could be used in operational settings to assist in HRB scheduling in the Sydney area, with the aim that burning could be avoided on days at high risk of hazardous $PM_{2.5}$ pollution. The models provide probabilistic predictions of mean and maximum Sydney daily $PM_{2.5}$ based on weather, fire area in different locations and yesterday's $PM_{2.5}$. The conditions most likely to produce hazardous $PM_{2.5}$ pollution are with low ventilation index in Sydney, low temperature, an afternoon sea breeze and westerly to northwesterly winds in the Blue Mountains, and large areas being burnt, particularly to the west and north of Sydney. Higher $PM_{2.5}$ the previous day also substantially increases predicted $PM_{2.5}$. The models were produced using Bayesian regression, which makes predictive outputs highly informative, including a clear communication of uncertainty, and flexible depending on user need. The models are fast-to-run and require easily accessible inputs, and could be used in conjunction with existing methods to improve information and decision-making around HRB scheduling.

Data of fires and $PM_{2.5}$ continues to be collected, thus future work could use new data to improve the precision of the models presented here. We focused on HRBs near Sydney, but a similar approach could be applied to develop a model applicable to wildfire prediction and applied to $PM_{2.5}$ and fire activity modelling in other regions of Australia and the world.

## Supporting information

**S1 Appendix. Effects plots for maximum Sydney PM2.5 model (Fig A) and additional tile plot of predictions for mean and maximum Sydney PM2.5 models (Fig B).**
(PDF)

**S2 Appendix. Method used to create VIIRS SNPP hotspot clusters.**
(PDF)

## Acknowledgments

The results contain modified Copernicus Climate Change Service information 2021. Neither the European Commission nor ECMWF is responsible for any use that may be made of the Copernicus information or data it contains.

Basemap in Fig 1 was created by OpenStreetMap contributors and made available under the Open Database License: https://www.openstreetmap.org/copyright

## Author Contributions

**Conceptualization:** Michael Anthony Storey, Owen F. Price.

**Data curation:** Michael Anthony Storey, Owen F. Price.

**Formal analysis:** Michael Anthony Storey.

**Funding acquisition:** Owen F. Price.

**Methodology:** Michael Anthony Storey, Owen F. Price.

**Project administration:** Owen F. Price.

**Supervision:** Owen F. Price.

**Writing – original draft:** Michael Anthony Storey.

**Writing – review & editing:** Owen F. Price.

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
