## [Decision Letter · Decision Letter 0]

12 Jul 2022

PONE-D-22-14315Prediction of air quality in Sydney, Australia as a function of forest fire load and weather using Bayesian statisticsPLOS ONE

Dear Dr. Storey,

Thank you for submitting your manuscript to PLOS ONE. After careful consideration, we feel that it has merit but does not fully meet PLOS ONE’s publication criteria as it currently stands. Therefore, we invite you to submit a revised version of the manuscript that addresses the points raised during the review process.

We look forward to receiving your revised manuscript.

Kind regards,

Peng Chen, Ph.D.

Academic Editor

PLOS ONE

Journal Requirements:

  "Michael Storey and Owen Price are a part of the NSW Bushfire Risk Management Research Hub, funded by the New South Wales Department of Planning, Industry and Environment (Australia)."  

    "This research was funded by the NSW Department of Planning, Industry and Environment, via the 

488 NSW Bushfire Risk Management Research Hub."

 "Michael Storey and Owen Price are a part of the NSW Bushfire Risk Management Research Hub, funded by the New South Wales Department of Planning, Industry and Environment (Australia)."

Reviewers' comments:

Reviewer's Responses to Questions

**Comments to the Author**

1. Is the manuscript technically sound, and do the data support the conclusions?

Reviewer #1: Yes

Reviewer #2: Yes

2. Has the statistical analysis been performed appropriately and rigorously? 

Reviewer #1: Yes

Reviewer #2: Yes

3. Have the authors made all data underlying the findings in their manuscript fully available?

Reviewer #1: Yes

Reviewer #2: Yes

4. Is the manuscript presented in an intelligible fashion and written in standard English?

Reviewer #1: Yes

Reviewer #2: Yes

5. Review Comments to the Author

Reviewer #1: This paper has achieved its primary purpose: to develop a model to assist in scheduling HRBs around Sydney to avoid or reduce HRB-induced smoking. This model is based on Bayesian statistics and can quickly produce probabilistic predictions with given regional fire and weather variables.

Please find a few insights to consider:

Maybe you could give some details about Bayesian Statistics and GAM, not just what packages are used.

The predicted r2 of the model given in this paper is about 0.5. Can you provide the accuracy of some other methods like physical-based model or the empirical model? It would be better to compare the runtime and accuracy with other methods.

Some other minor things to check:

Line175: "We calculated daily mean values for all variables used in modelling, expect for the coastal wind U..." "expect" may be "except"

Line218: r2 needs to be explained.

Reviewer #2: General comment

The manuscript discusses the development of a probabilistic model of daily PM2.5 across the Sydney area using Bayesian regression. The model provides a general, fast-to-run and operationally useful assessment of daily PM2.5 levels in the Sydney basin as a function of regional fire and weather variables. The manuscript is interesting and worth publication. It can be accepted after modification.

Detail Comment

1. The abstract is not very well written and represents the overall information in the manuscript. I suggest the authors improved their abstract with a clear and balance problem statement, main objective, main finding, conclusion, and suggestions.

2. Line 28: “The model is expected….”. Another sentence that needs more concrete information based on the output of this study.

3. I suggest the authors check all the references in the main text and make sure they were written based on the journal reference style. For example, in the first paragraph F. H. Johnston, 2012 and Fay H. Johnston et al., 2021 should be written as Johnston et al., 2012 and Johnston et al., 2021, respectively.

4. I would like to suggest the authors improve the flow of information and paragraphing for their introduction. Make sure each paragraph consists of one point of information.

5. The first few sentences about Sydney in paragraph two can be included in the methodology.

6. The title of the manuscript is “prediction of air quality” but is any particular reason why only PM2.5 was emphasised?

7. Results; “Raw Data Summary” – Descriptive Analysis Results? How about fire data? Fire data were mentioned in the “Methodology”.

8. Section 4.2. “Model Results” seems not a suitable title for this section. I suggest the authors include a more complete title to represent the content of the section.

9. Line 358-359: The authors are suggested to include their suggestions for future studies in the Conclusion.

6. PLOS authors have the option to publish the peer review history of their article (what does this mean?). If published, this will include your full peer review and any attached files.

Reviewer #1: No

Reviewer #2: No

---

## [Author Response · Author response to Decision Letter 0]

26 Jul 2022

See our responses to journal requirements in the cover letter and our responses to reviewers in the document named "Responses to Reviewers"

---

## [Editor Report · Decision Letter 1]

27 Jul 2022

Prediction of air quality in Sydney, Australia as a function of forest fire load and weather using Bayesian statistics

PONE-D-22-14315R1

Dear Dr. Storey,

We’re pleased to inform you that your manuscript has been judged scientifically suitable for publication and will be formally accepted for publication once it meets all outstanding technical requirements.

Kind regards,

Peng Chen, Ph.D.

Academic Editor

PLOS ONE
---

## [Editor Report · Acceptance letter]

1 Aug 2022

PONE-D-22-14315R1 

Prediction of air quality in Sydney, Australia as a function of forest fire load and weather using Bayesian statistics 

Dear Dr. Storey:

I'm pleased to inform you that your manuscript has been deemed suitable for publication in PLOS ONE. Congratulations! Your manuscript is now with our production department. 

Kind regards, 

on behalf of

Dr. Peng Chen 

Academic Editor

PLOS ONE